# Pharmacodynamics of Doripenem Alone and in Combination with Relebactam in an In Vitro Hollow-Fiber Dynamic Model: Emergence of Resistance of Carbapenemase-Producing *Klebsiella pneumoniae* and the Inoculum Effect

**DOI:** 10.3390/antibiotics12121705

**Published:** 2023-12-07

**Authors:** Elena N. Strukova, Maria V. Golikova, Svetlana A. Dovzhenko, Mikhail B. Kobrin, Stephen H. Zinner

**Affiliations:** 1Department of Pharmacokinetics & Pharmacodynamics, Gause Institute of New Antibiotics, 11 Bolshaya Pirogovskaya Street, 119021 Moscow, Russia; strukovagause@gmail.com (E.N.S.); sad35@yandex.ru (S.A.D.); mbkobrin@gmail.com (M.B.K.); 2Harvard Medical School, Department of Medicine, Mount Auburn Hospital, 330 Mount Auburn St., Cambridge, MA 02138, USA; szinner@mah.harvard.edu

**Keywords:** antibiotic resistance, inoculum effect, in vitro hollow-fiber dynamic model, beta-lactams, beta-lactamase inhibitors, doripenem, relebactam, *Klebsiella pneumoniae*, LC-MS/MS

## Abstract

The emergence of bacteria resistant to beta-lactam/beta-lactamase inhibitor combinations is insufficiently studied, wherein the role of the inoculum effect (IE) in decreased efficacy is unclear. To address these issues, 5-day treatments with doripenem and doripenem/relebactam combination at different ratios of the agents were simulated in a hollow-fiber dynamic model against carbapenemase-producing *K. pneumoniae* at standard and high inocula. Minimal inhibitory concentrations (MICs) of doripenem alone and in the presence of relebactam at two inocula were determined. Combination MICs were tested using traditional (fixed relebactam concentration) and pharmacokinetic-based approach (fixed doripenem-to-relebactam concentration ratio equal to the therapeutic 24-h area under the concentration-time curve (AUC) ratio). In all experiments, resistant subpopulations were noted, but combined simulations reduced their numbers. With doripenem, the IE was apparent for both *K. pneumoniae* isolates in combined treatments for one strain. The pharmacokinetic-based approach to combination MIC estimation compared to traditional showed stronger correlation between DOSE/MIC and emergence of resistance. These results support (1) the constraint of relebactam combined with doripenem against the emergence of resistance and IE; (2) the applicability of a pharmacokinetic-based approach to estimate carbapenem MICs in the presence of an inhibitor to predict the IE and to describe the patterns of resistance occurrence.

## 1. Introduction

Beta-lactamase-producing *Klebsiella pneumoniae* have become a global emergency. Novel antimicrobial combinations of beta-lactam and beta-lactamase inhibitors such as imipenem/relebactam have been approved to treat infections caused by such bacterial pathogens [1]. Numerous in vitro studies have demonstrated activity of this combination; however, resistant strains are beginning to appear in hospitals [2,3]. In addition to the growing emergence of resistance, diminished efficacy of antibiotics at high bacterial inocula known as the inoculum effect (IE) also might play an important role in decreased efficacy against antibiotic-susceptible bacterial strains. The IE of beta-lactams and beta-lactam/beta-lactamase inhibitor combinations is often detected under in vitro conditions at MIC testing of bacterial pathogens. However, the clinical relevance of the IE is not clear. Some in vivo studies did not confirm diminished efficacy of beta-lactams and beta-lactam/beta-lactamase inhibitor combinations at higher bacterial inocula, [4,5,6,7,8] while others suggested that the in vitro inoculum effect correlates well with in vivo outcomes in animal models [9,10].

The in vitro hollow-fiber infection model (HFIM) is an effective tool to study antimicrobial efficacy; it allows tracking of the emergence of bacterial resistance and estimation of the potential clinical utility of the IE, as it enables the simulation of human antibiotic pharmacokinetics in plasma or at the infection site as well as the evaluation of the pharmacodynamics of simulated clinical antibiotic courses [11,12]. These models are recommended by the European Medicines Agency to describe pharmacokinetic-pharmacodynamic (PK/PD) relationships for representative organisms over a wide range of inocula [13]. HFIM is used in the current pharmacodynamic (PD) study to evaluate bacterial resistance and the IE.

According to a traditional approach, antibiotic MICs in the presence of an inhibitor are assessed using a fixed concentration of the inhibitor [14]. However, during therapy the concentrations of both the inhibitor and the antibiotic are continuously changing. To account for these changes a pharmacokinetic-based (PK-based) approach to MIC determination of beta-lactam/beta-lactamase inhibitor combinations has been proposed recently [15]. The antibacterial effects of imipenem/relebactam and doripenem/relebactam combinations in time-kill experiments were accurately predicted using this approach, i.e., with MICs determined using a PK-based antibiotic-to-inhibitor concentration ratio.

The first goal of the present study is using HFIM to monitor the emergence of doripenem resistance in *Klebsiella pneumoniae* carbapenemase (KPC)-producing *K. pneumoniae* during simulations of different regimens of doripenem alone or combined with the beta-lactamase inhibitor relebactam [16]. The second goal is to study the influence of *K. pneumoniae* inocula density on the effect of doripenem and doripenem combined with relebactam. To address these goals, therapeutic, sub-therapeutic, and supra-therapeutic exposures of doripenem and relebactam in 5-day single and combined treatments were simulated in an in vitro hollow-fiber dynamic model. The third goal is to compare the predictive potential of MICs estimated by both the traditional and the novel PK-based approaches (at standard and high-density inocula) with respect to the antibacterial effect of doripenem in combination with relebactam and the emergence of resistant subpopulations of KPC-producing *K. pneumoniae*.

## 2. Results

### 2.1. MICs of Doripenem Alone and in the Presence of Relebactam

The MICs of doripenem for *K. pneumoniae* 19 and *K. pneumoniae* 28 at two inocula differed 16–32-fold, so the IE was obvious (Table 1). Similar observations were made for doripenem in the presence of a fixed relebactam concentration of 4 mg/L assessed by method 1: combination MIC1s at a high inoculum (HI) compared to a standard inoculum (SI) were reduced 64-fold. By method 2, in the presence of relebactam the difference in carbapenem susceptibility of tested *K. pneumoniae* strains depended on the strain, as well as on the doripenem-to-relebactam concentration ratio and the inocula. At a 1.5/1 concentration ratio that corresponds to the therapeutic doripenem-to-relebactam AUC_24_ ratio, the IE was observed for the more doripenem susceptible *K. pneumoniae* 19 as the MIC2 of doripenem increased 8-fold while for *K. pneumoniae* 28 the increase was 4-fold. Therefore, the doripenem/relebactam combination was prone to the IE for one of the two tested strains. A similar observation was made with a 6/1 doripenem-to-relebactam concentration ratio: the IE was observed only for *K. pneumoniae* 19 (16-fold MIC2 increase), while for *K. pneumoniae* 28 doripenem MIC2 at HI increased only 2-fold. With doripenem-to-relebactam combination at a ratio of 1/3, IE was observed for both *K. pneumoniae* strains.

### 2.2. Doripenem and Doripenem/Relebactam Combination Pharmacokinetics

At SI, with both *K. pneumoniae* isolates the determined concentrations of doripenem alone were consistent with the targeted values only for the first infusion (Figure 1, data from the experiments with *K. pneumoniae* 19 and therapeutic simulations shown as an example). On the second day of the experiment, antibiotic concentrations were significantly lower than expected, due to doripenem hydrolysis by bacterial carbapenemases. At HI, doripenem concentrations were always below the targeted values for both isolates. In combined treatments, doripenem and relebactam concentrations for both *K. pneumoniae* isolates at either SI or HI reached the targeted values.

### 2.3. Doripenem and Doripenem/Relebactam Combination Pharmacodynamics with K. pneumoniae Total and Resistant Subpopulations

We chose the relevant clinical situation of high-burden bacterial infections to study doripenem and doripenem/relebactam pharmacodynamics and the emergence of resistance. We conducted 5-day PD experiments with a starting bacterial concentration of ~10^8^ CFU/mL in the bioreactor. Doripenem regimens were simulated to mimic therapeutic AUC_24_ after a 500 mg 0.5-h infusion (120 mg × h/L—regimen 120D), half of the therapeutic AUC_24_ (60 mg × h/L—regimen 60D), and twice the therapeutic AUC_24_ (240 mg × h/L—regimen 240D). With the doripenem/relebactam combination, different AUC_24_ ratios of doripenem and relebactam were simulated: (i) at therapeutic AUC_24_s of both drugs (120D and 80 mg × h/L for relebactam—80R, this regimen was designated as 120D/80R and corresponds to a 1.5/1 AUC_24_ drug ratio), (ii) at twice doripenem and half relebactam therapeutic AUC_24_s (regimen 240D/40R, 6/1 AUC_24_ ratio), and (iii) at half doripenem and twice relebactam therapeutic AUC_24_s (regimen 60D/160R, 1/2.7 rounded to 1/3 AUC_24_ ratio). Monitoring of doripenem-resistant subpopulations of *K. pneumoniae* was achieved by daily plating samples with bacteria on agar plates with doripenem at concentrations ranging from 2–16× MIC. The numbers of resistant variants that grew on the plates with 4× MICof doripenem was used to describe the selection of doripenem resistance; the data for plates with 2, 8, and 16× MIC of doripenem were consistent with 4× MIC data.

Results of these experiments are shown in Figure 2. Single doripenem simulations (Figure 2, higher panel, a–c) of 60D and 120D showed no antibacterial effect against total population (agar plates without antibiotic) with both *K. pneumoniae* strains. At regimen 240D, weak antibacterial effect with regrowth at 24 (*K*. *pneumoniae* 28) and 48 h (*K. pneumoniae* 19) to 8.5 log CFU/mL was observed.

At 60D regimen, subpopulations of *K. pneumoniae* 28 resistant to 4× MIC of doripenem stayed close to the limit of detection during the entire observation period. The numbers of doripenem-resistant cells of *K. pneumoniae* 19 grew by 1–1.5 log CFU/mL (Figure 2g–i); however, even less than at therapeutic AUC_24_ (plateau at 4–5 log CFU/mL after 24 h). At twice-the therapeutic regimen (240D), subpopulations resistant to 4× MIC of doripenem reached 8.5 log CFU/mL as was the total population; growth of *K. pneumoniae* 28 began immediately after starting the doripenem infusion, while resistant cells of *K. pneumoniae* 19 grew at a 24 h delay.

Simulation of the therapeutic combined treatment 120D/80R led to a similar decrease in total bacterial numbers with both strains (3–4 orders), but with slightly lower bacterial concentrations of *K. pneumoniae* 19 followed by gradual regrowth after 24 h, although the total population did not reach the starting inoculum (Figure 2d–f). Similar trends in the total bacterial population growth kinetics were observed with D240/R40 and 60D/160R combination regimens.

The growth of doripenem-resistant subpopulations of both *K. pneumoniae* strains was suppressed by all combined treatments compared with doripenem alone (Figure 2, lower panel, j–l). However, this suppression was incomplete. At the therapeutic regimen 120D/80R, subpopulations of *K. pneumoniae* 19 resistant to 4× MIC of doripenem were reduced under the limit of detection until 96 h when slight regrowth began; with *K. pneumoniae* 28, regrowth after 24 h was accelerated and reached 6.7 log CFU/mL by the end of the observation period. At the 240D/40R combination regimen, resistant subpopulations of both strains decreased after the start of simulation, and modest regrowth of the more susceptible strain *K. pneumoniae* 19 appeared only after 96 h (3.4 log CFU/mL), while with *K. pneumoniae* 28, resistant cells increased after 24 h and gradually reached 6.3 log CFU/mL. In combined regimen 60D/160R, regrowth of resistant subpopulations of *K. pneumoniae* 28 was delayed until 96 h, while *K. pneumoniae* 19 reached 3.4 log CFU/mL by 48 h; however, both strains showed ~3 log CFU/mL by the end of the observation period. In general, in all combined treatments the more doripenem-resistant strain *K. pneumoniae* 28 showed a weaker antimicrobial effect that was accompanied by enhanced growth of resistant subpopulations. In addition, among the simulated combination treatments, the most effective suppression of doripenem-resistant mutants was observed with the regimen in which relebactam prevails, i.e., 60D/160R.

Simulations of 60D with both strains and 120D with *K. pneumoniae* 28 produced no change in susceptibility at 120 h; with all other regimens of doripenem alone, MICs at the end of the simulations were 128 mg/L, which represents a 32- and 8-fold difference for *K. pneumoniae* 19 and *K. pneumoniae* 28, respectively. Combination simulations for *K. pneumoniae* 19 resulted in a 4–8-fold MIC change for 120D/80R and 240D/40R and a 4-fold MIC change for 60D/160R. With *K. pneumoniae* 28, MICs increased 4–8-fold at the end of all combined simulations.

### 2.4. Doripenem and Doripenem/Relebactam Combination IE at Therapeutic Regimens

MIC testing revealed the IE for doripenem alone with both *K. pneumoniae* strains. For the doripenem/relebactam combination at a drug concentration ratio of 1.5/1, which corresponds to the therapeutic doripenem-to-relebactam AUC_24_ ratio, the IE was detected only with *K. pneumoniae* 19 (8-fold inocula-related MIC shift), but not with the more doripenem-resistant strain (4-fold inocula-related MIC shift). At the same time, using the traditional approach with fixed relebactam content, a prominent IE was observed for both *K. pneumoniae* strains (32–64-fold inocula-related MIC shift). To investigate the IE occurrence with therapeutic regimens of doripenem and doripenem/relebactam combinations, PD experiments with SI were conducted in HFIM. Visual comparisons of the time courses of *K. pneumoniae* at SI and HI were plotted on the same graph (Figure 3).

As seen in Figure 3a,b, in experiments at SI, within the first 6 h of treatment, doripenem effectively reduced initial cell numbers by 2–2.7 log CFU/mL depending on the strain. However, bacterial killing was followed by intensive regrowth, and within the first 24–48 h bacterial counts increased to 9 log CFU/mL. In combination treatments (Figure 3c,d), at both inocula doripenem displayed more pronounced antimicrobial effects compared with monotherapy. As seen in the figure, the difference between bacterial counts for both strains at standard and high inocula exposed to doripenem alone was noticeable over the first 24–48 h; exposure to doripenem/relebactam showed an obvious difference only for *K. pneumoniae* 19. Therefore, at SI minimal bacterial counts were equal to the limit of detection at 48 h and then cell numbers gradually increased to 7 log CFU/mL by the end of the observation. When the starting bacterial load was high, the effect of doripenem/relebactam was less pronounced; after 24 h of doripenem exposure when minimal numbers reached 3 log CFU/mL, regrowth was observed. Therefore, with *K. pneumoniae* 19 the efficacy of doripenem/relebactam was clearly influenced by the bacterial concentration at the start of treatment; similarly, this was observed for both strains with doripenem alone but to a lesser extent. With *K. pneumoniae* 28 at SI in the presence of relebactam, doripenem lowered bacterial counts similar to HI, and during most of the observation period, the course of bacterial counts at the two inocula was generally similar. These data suggest that the effect of the doripenem/relebactam combination was independent of the starting inoculum of *K. pneumoniae* 28 and the IE was not detected.

### 2.5. Predictive Potential of DOSE/MIC with MICs Estimated by Method 1 and Method 2 in Terms of Emergence of Resistance

To analyze the relationships between the emergence of doripenem-resistant *K. pneumoniae* (AUBC_M_) in mono and combined treatments at HI and simulated doripenem DOSE/MIC ratios, MIC data obtained at SI and HI were used. To calculate the DOSE/MIC ratios for combination treatments, MICs determined by the traditional (method 1) and PK-based (method 2) approaches were used. As a result, two DOSE/MIC data sets were obtained for each inocula density (SI and IH): for the first, DOSE/MIC ratios were based on MICs of doripenem alone and in combination with relebactam determined by method 1; for the second, DOSE/MIC ratios were based on MICs of doripenem alone and in combination with relebactam determined by method 2. Therefore, two “DOSE/MIC–AUBC_M_” relationships at each inocula density were constructed; this allowed the comparison of the predictive potential of combination MICs at SI and HI determined by the two methods (Figure 4).

When the MICs determined at SI were used to construct the “DOSE/MIC–AUBC_M_” relationships, independently of the method used to determine the doripenem/relebactam combination MICs, the data sets were poorly described with a Gaussian function (*r*^2^ did not exceed 0.31). In contrast, the use of HI MICs allowed a Gaussian function to describe both variants of the relationship. Stronger correlation was observed when MICs in DOSE/MIC parameter were calculated using the PK-based approach (method 2)—squared correlation coefficients were 0.84 (PK-based approach) and 0.60 (method 1).

## 3. Discussion

This study was designed to investigate the emergence of resistance of *K. pneumoniae* to doripenem alone and combined with the carbapenemase inhibitor relebactam at different antibiotic and inhibitor ratios in HFIM. Bacterial inocula at the start of simulations approximated 10^8^ CFU/mL (HI) as a clinically relevant scenario. To assess the significance of the IE, additional PD experiments were designed with starting inocula of 10^6^ CFU/mL exposed to doripenem alone or in combination with relebactam at therapeutic regimens.

With all simulated regimens, resistant subpopulations were not fully suppressed, although much weaker growth was seen with the combination. Even at the highest doripenem monotherapy regimen (twice therapeutic AUC_24_, 240D), abundant growth of resistant subpopulations was seen (Figure 2). More active and faster growth of resistant subpopulations was seen with the more resistant *K. pneumoniae* 28 strain. With combination treatments, the most effective suppression of resistance was seen with regimen 60D/160R with a greater presence of the inhibitor; growth of resistant *K. pneumoniae* also was restrained with the two other combination regimens, but slight regrowth was observed. According to these data, the relebactam should be added to doripenem in excess to decrease the emergence of resistance and increase treatment efficacy.

Doripenem exposure (expressed as DOSE/MIC ratio) and resistance data obtained from the PD experiments, merged for mono and combined treatments, were used to establish “DOSE/MIC–AUBC_M_” relationships. To calculate DOSE/MIC ratios for combined treatments, doripenem/relebactam combination MICs were determined by method 1 or method 2. All MICs were determined at SI and HI to check which MIC more adequately reflects the antibiotic/inhibitor combination antimicrobial potential. Four “DOSE/MIC–AUBC_M_” relationships were constructed: two for SI (Figure 4a,b) and two for HI (Figure 4c,d). As a result, bell-shaped relationships described by a Gaussian function could be established only when MICs were determined at HI.

It was unexpected that standard MICs at SI did not establish strong enough relationships of resistance emergence with doripenem exposure. As reported in numerous studies, MIC-based PK/PD parameters such as AUC/MIC and T_>MIC_ have been used successfully to establish “exposure–resistance” relationships for antibiotics of different classes [17,18,19,20,21,22,23]. Such a relationship for beta-lactamase producing *K. pneumoniae* exposed to meropenem (DOSE/MIC with MICs determined at SI) has been described by our laboratory [24]. It is possible that combining data for mono and combined treatments in the current study poses some difficulties. On the other hand, it seems logical that MICs determined at HI provide good correlation with anti-mutant efficacy, since PD experiments were conducted under similar conditions of a dense bacterial load.

Based on the “DOSE/MIC-AUBC_M_” relationship (Figure 4d), AUBC_M_s corresponding to the 240D regimen for both strains fall at the top of the bell-shaped curve. Further increasing the DOSE/MIC ratio and decreasing the MIC by potentiating doripenem with relebactam could effectively suppress resistant subpopulations.

In the current study, the results of MIC testing revealed the IE for doripenem alone. With the combination MICs, the difference between SI and HI was also detected, but it was most pronounced when method 1 with a fixed relebactam concentration was used. At HI, the amount of relebactam was insufficient to keep the doripenem MIC at the same level as at SI, since the relebactam concentration was equal at both inocula (4 mg/L). With method 2, doripenem and relebactam concentrations are identical and combination MIC data at the two inocula do not differ as much as with method 1. The results of combination MIC testing with method 2 showed that combination activity at HI was less restricted with the relative predominance of doripenem. Moreover, the IE was detected for the more susceptible *K. pneumoniae* 19. With the predominance of relebactam combination, MICs at SI for both *K. pneumoniae* strains were noticeably lower than with the prevalence of doripenem (0.25–0.5 mg/L at 1/3 ratio vs. 1–8 mg/L at 1.5/1 and 6/1 doripenem/relebactam ratios). Obviously, a carbapenemase inhibitor plays an important role in the antimicrobial activity against carbapenemase producing bacteria, so when its presence was adequate, the combination was more active. However, the activity of combination 1/3 was diminished at HI and the IE was observed with both studied strains. The tendency for a greater MIC difference between SI and HI with the more susceptible *K. pneumoniae* strain was observed in all but one of the studied combinations (with a fixed relebactam concentration), similar to our previous study [15]. It is possible that carbapenemase enzymes are more active at increased inocula against susceptible rather than highly resistant strains. Resistant strains produce high concentrations of beta-lactamases, so the difference between HI and SI may not be easily discriminated.

To support this explanation, we conducted a series of PD simulations at two inocula (SI and HI) with therapeutic doripenem alone and in combination with relebactam (1.5/1 doripenem-to-relebactam AUC_24_ ratio). In these experiments, the IE was observed for both strains with doripenem alone, but in combined treatments the IE was detected only with the more susceptible strain, *K. pneumoniae* 19. Notably, the PK-based approach for MIC determination provided a more realistic estimation of both the antibiotic/inhibitor antimicrobial activity and the IE.

The results of our study indicate the usefulness of the PK-based approach to estimate carbapenem MICs in the presence of an inhibitor and to establish the relationship with the emergence of resistance. V. Tam and colleagues have reported that traditional MIC estimation for beta-lactam/beta-lactamase inhibitor combinations may not be optimal to guide dosing regimens to combat regrowth of resistant subpopulations. These authors used time above MIC to assess the antibacterial effect of piperacillin and ceftazidime in combination with tazobactam or avibactam against *K. pneumoniae* and *Escherichia coli* [25,26]. A T_˃MICi_ > 73.6 or 76% should suppress regrowth with HI in HFIM [25,26]. These authors mention that for ceftazidime in combination with avibactam, the breakpoint value of T_˃MICi_ in HFIM depended on a cell density for SI T_˃MICi_ ≥ 55% was sufficient to suppress the regrowth while HI demanded ≥ 73.6 [25].

Our study has several limitations. First, it did not include a large number of *K. pneumoniae* strains, particularly non-carbapenemase producers, to verify the role of various resistance mechanisms. Additional PD studies with *K. pneumoniae* isolates and other species of Gram-negative bacteria are necessary to more completely evaluate the applicability of the PK-based approach to MIC testing and predicting the emergence of resistance and the IE. Subsequent studies with a wide range of bacteria and beta-lactam/beta-lactamase inhibitor combinations would enhance the generalizability of our results.

## 4. Materials and Methods

Methods used to produce the dataset are shown in the flowchart (Appendix A).

### 4.1. Antimicrobial Agents and Bacterial Strains

Doripenem hydrate powder was purchased from Acros Organics (Fair Lawn, NJ, USA). Relebactam was purchased from Invivochem (Libertyville, IL, USA). Two bla KPC positive by PCR clinical *K. pneumoniae* isolates differing in doripenem susceptibility were used in this study: *K. pneumoniae* 28 and 19; *K. pneumoniae* ATCC 700603 was used as a negative control. Before each testing, carbapenemase production was confirmed for each bacterial strain by a modified carbapenem-inactivation method [27].

### 4.2. Susceptibility Testing

Susceptibility testing for antibiotic and inhibitor alone or in combination was performed using broth microdilution techniques with a standard inoculum of approximately 5 × 10^5^ CFU/mL (SI) and high inoculum—5 × 10^7^ CFU/mL (HI). Single doripenem MICs at SI were determined according to the standard recommendations using Mueller-Hinton broth (Becton Dickinson, Franklin Lakes, NJ, USA) [28]. When the MICs (for single doripenem and for its combination with relebactam) were determined at HI, bacterial growth was quantified by optical density at 600 nm (OD), and ODs before and after 18 h incubation at 37 °C were estimated. The MIC was defined as the dilution at which the 18 h OD was equal to or less than that at time 0. IE was defined as an eightfold or greater increase in MIC when tested with HI compared to that for SI. For doripenem/relebactam combinations, MIC testing was performed by method 1 or method 2 regarding the ratio of doripenem to relebactam. Before reading, plates were incubated at 37 °C for 18 h. MIC values were obtained at least in triplicate, and the modal MICs were estimated.

Method 1 (standard, MIC1). MIC testing for doripenem/relebactam combination used a fixed relebactam concentration of 4 mg/L with doubling dilutions of doripenem.

Method 2 (PK-based, MIC2). MIC testing for doripenem/relebactam combinations used a fixed PK-based carbapenem-to-relebactam concentration ratio of 1.5/1, 6/1, and 1/3 by varying the carbapenem and relebactam concentrations in parallel in each subsequent dilution. These concentration ratios are equal to the 24-h area under the concentration-time curve (AUC_24_) ratios of doripenem to relebactam simulated in HFIM. A 1.5/1 doripenem-to-relebactam concentration ratio in MIC testing corresponds to the therapeutic 1.5/1 AUC_24_ ratio of doripenem (for a 500 mg dose of every 8 h) to relebactam (for a 250 mg dose every 8 h): 120 mg × h/L of doripenem to 80 mg × h/L of relebactam, 120/80 = 1.5/1. A 6/1 doripenem-to-relebactam concentration ratio in MIC testing corresponds to the 6/1 ratio of twice-therapeutic AUC_24_ of doripenem to half-therapeutic AUC_24_ of relebactam: 240 mg × h/L of doripenem to 40 mg × h/L of relebactam, 240/40 = 6/1. A 1/3 doripenem-to-relebactam concentration ratio in MIC testing corresponds to the 1/3 ratio of half-therapeutic AUC_24_ of doripenem to twice-therapeutic AUC_24_ of relebactam: 60 mg × h/L of doripenem to 180 mg × h/L of relebactam, 60/160 = 1/2.7 was rounded up to 1/3.

### 4.3. Antibiotic Dosing Regimens and Simulated Pharmacokinetic Profiles

Simulated plasma AUC_24_ of doripenem (120 mg × h/L) both in single and combined treatments with relebactam corresponded to the antibiotic therapeutic dosing regimen of 500 mg as a 0.5-h infusion every 8 h [29]. Relebactam AUC_24_ (80 mg × h/L) in combined treatments with doripenem was calculated using the peak serum inhibitor concentration and t1/2 reported in human studies with a 0.5-h infusion of 250 mg of relebactam [30]. Additional regimens of half and twice plasma AUC_24_ of doripenem in single (60D and 240D) and combined treatments with half and twice plasma AUC_24_ of relebactam at HI were simulated (240D/40R and 60D/160R regimens). With all dosing regimens, a series of monoexponential profiles that mimic trice-daily dosing of doripenem and relebactam with a half-life for both drugs of 1.2 h used alone or in combination were simulated for 5 consecutive days.

### 4.4. In Vitro Dynamic Model

A previously described two-compartment in vitro model (a hollow-fiber infection model) [11] was used in PD simulations of single and combined treatments with doripenem and relebactam. Briefly, the model consists of three connected cameras, one containing fresh cation supplemented Mueller–Hinton broth (CSMHB), this supplies with CSMHB the second camera–central unit that is used for drug dosing, and the third camera–hollow-fiber bioreactor (Fresenius dialyzer, model AV400S, Fresenius Medical Care AG, Bad Homburg Germany) is a peripheral unit that is used for bacterial cultivation and represents the infection site. The central unit and bioreactor are connected, and continuous exchange of CSMHB between these units by peristaltic pump provides maintenance of target drug concentrations in both cameras.

The operational procedure used in the PD experiments was as described elsewhere [24]. Each experiment was performed at least in duplicate. Antibiotic dosing and sampling were processed automatically using computer-assisted controls. The system was filled with sterile CSMHB and placed in an incubator at 37 °C. The hollow-fiber bioreactor was inoculated with an 18 h culture of *K. pneumoniae* at appropriate cell concentration. After a 2 h period of incubation, the resulting exponentially growing cultures reached ~5 × 10^5^–10^6^ (SI) or ~5 × 10^7^–10^8^ (HI) colony-forming units (CFU)/mL. Then, antibiotic or antibiotic with inhibitor were administered into the central unit of the model. The duration of each experiment was 120 h.

To verify the reliability of pharmacokinetic simulations, throughout each experiment the bioreactor was multiply sampled immediately after the end of infusion and at the end of the dosing interval (6 h). Doripenem and relebactam concentrations were determined by an acetonitrile protein precipitation and liquid chromatography-tandem mass spectrometry (LC-MS/MS) according to the method proposed earlier for the determination of relebactam [30]. Meropenem was used as an internal standard (IS). The system consisted of an Ultimate 3000 liquid chromatograph and a TSQ Quantum Access MAX triple-quadrupole tandem mass spectrometer (Thermo Scientific, Waltham, MA, USA) equipped with an electrospray ionization source operating in the positive mode with 5 kV spray voltage. The temperature of the capillary transfer and vaporizer were set at 100 and 220 °C, respectively. The nitrogen flow rate employed as sheath gas was set at 20 arbitrary units. Argon was used as collision gas (1.4 mTorr) and collision energy was set at 16 eV. The tube lens was set at 164, 159, and 142 V for relebactam, doripenem, and meropenem, respectively. The analysis was monitored on SRM mode. The transition for relebactam [M+H]^+^: *m*/*z* 349,1 → 269,1, for doripenem [M+H]^+^: *m*/*z* 421 → 274, and for IS (meropenem) [M+H]^+^: *m*/*z* 384 → 141. The chromatographic separation of the analytes (5 µL) was achieved using a mobile phase of 5 mM ammonium acetate (pH 4.5) in 80:20 acetonitrile-water. The flow rate and run time were 0.15 mL/min and 7 min, respectively. Sample preparation: 50 µL of broth with tested drugs were placed in a 1.5 mL eppendorf plastic tube, 150 µL of water with meropenem at a concentration of 7.33 µg/mL were added and shaken for 1 min, and then 800 µL of acetonitrile were added and centrifuged for 5 min at 13,000 rpm. The supernatant (5 µL) was analyzed by LC-MS/MS.

### 4.5. Quantitation of the Antimicrobial Effect and the Emergence of Resistance

In each experiment, bacteria-containing medium from the hollow-fiber bioreactor was sampled to determine bacterial, antibiotic, and inhibitor concentrations throughout the observation period. Samples (100 µL) were serially diluted as appropriate and 100 µL was plated onto Mueller–Hinton agar plates, containing 0×, 2×, 4×, 8×, and 16× MIC of doripenem, and plates were placed in an incubator at 37 °C for 24–48 h. The lower limit of accurate detection of susceptible cells was 1 × 10^2^ CFU/mL (equivalent to 10 colonies per plate) and of resistant cells this was 1 × 10^1^ CFU/mL (equivalent to at least one colony per plate).

Based on growth curves of *K. pneumoniae* subpopulations resistant to 4× MIC of doripenem, in each case area under the bacterial mutant concentration-time curve (AUBC_M_) [31] was determined from the beginning of treatment to 120 h.

Observed doripenem concentrations in mono treatment simulations were noticeably lower than the desired levels; therefore, indices such as AUC were negligible or close to zero. Thus, the Dose/MIC index was used to establish “concentration–response” relationships as was proposed in our previous study with meropenem [24]. Relationships between DOSE/MIC (using MIC1 and MIC2) and emergence of resistance (AUBC_M_) were described with the Gaussian (Equation (1)) function:*Y* = *a* exp [0.5 (*x* − *x*_0_)^2^/*b*](1)
where *Y* is AUBC_M_, *x* is log (DOSE/MIC), *x*_0_ is log (DOSE/MIC) that corresponds to the maximal value of *Y*, *a* is the maximal value of *Y*, and *b* is a parameter.

## 5. Conclusions

These results revealed the following: (1) the inability of therapeutic regimens of doripenem as well as doripenem plus relebactam to completely suppress the emergence of resistance; (2) the possible clinical significance of the IE for carbapenems without an inhibitor as well as strain-dependent occurrence of the IE with doripenem plus relebactam; (3) the applicability of a pharmacokinetic-based approach to estimate carbapenem MICs in the presence of an inhibitor to establish the relationship between the emergence of resistance and a PK/PD parameter–DOSE/MIC.

## Figures and Tables

**Figure 1 antibiotics-12-01705-f001:**
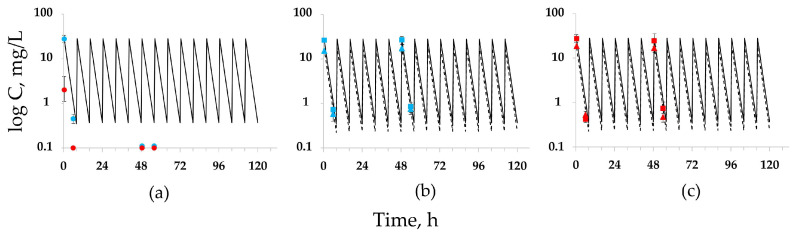
Simulated therapeutic profiles and observed concentrations of doripenem and relebactam. Targeted (line—doripenem, dotted line—relebactam) and determined (mean ± SD) concentrations of doripenem at SI (blue circles) and HI (red circles) in monotreatments (**a**), doripenem (squares), and relebactam (triangles) in combined treatments at SI (**b**) and HI (**c**).

**Figure 2 antibiotics-12-01705-f002:**
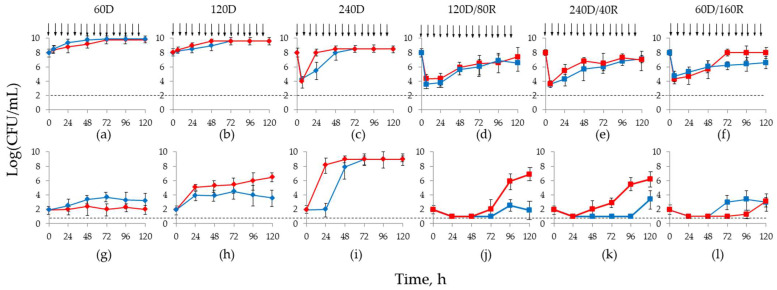
Time-kill curves of total bacterial population (**a**–**f**) and growth curves of doripenem-resistant subpopulations (4× MIC) (**g**–**l**) of *K. pneumoniae* 19 (blue symbols) and 28 (red symbols) in PD simulations at HI. Simulated regimens are indicated at the top of the figure as alphanumeric designations. Arrows indicate the start of doripenem or doripenem and relebactam infusion. Dotted lines indicate the limit of detection.

**Figure 3 antibiotics-12-01705-f003:**
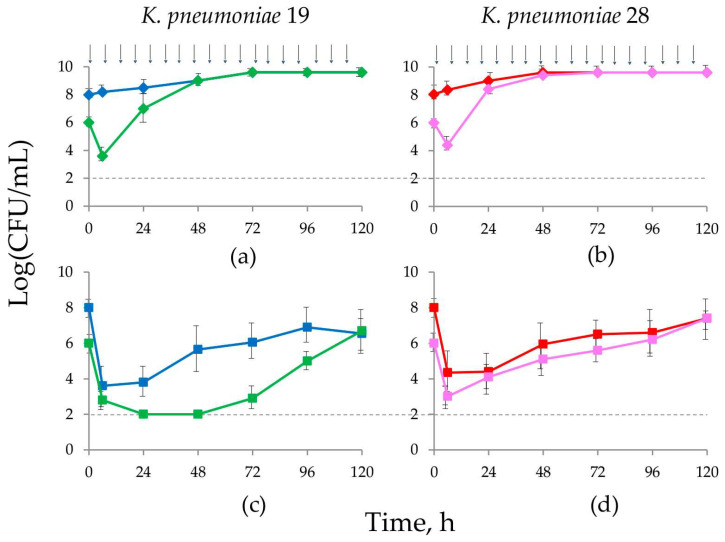
Time-kill curves of therapeutic AUC_24_s of doripenem (**a**,**b**) and doripenem/relebactam combination (**c**,**d**) against *K. pneumoniae* 19 and *K. pneumoniae* 28 obtained at SI (green and pink symbols) and HI (blue and red symbols). Arrows indicate the start of doripenem infusion. Dotted lines indicate the limit of detection.

**Figure 4 antibiotics-12-01705-f004:**
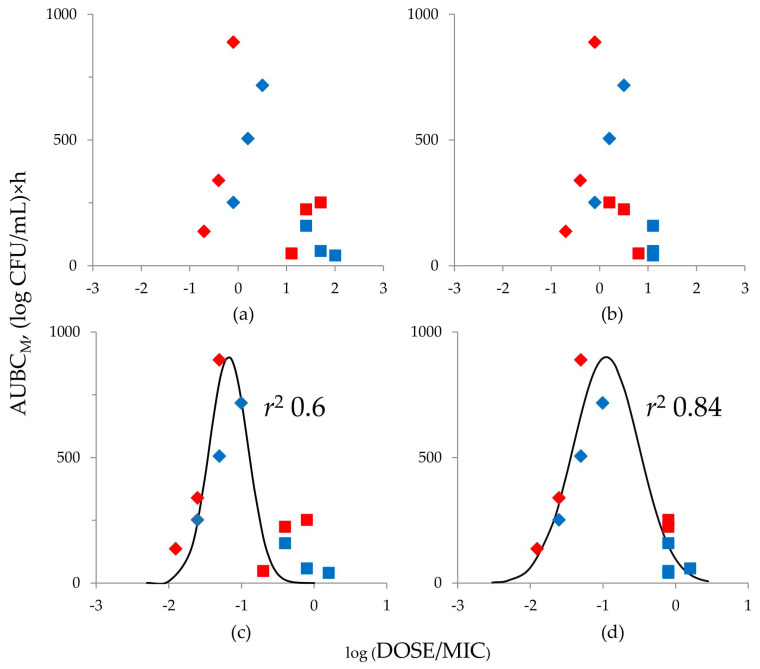
DOSE/MIC relationships with the emergence of doripenem-resistant *K. pneumoniae* 19 (blue symbols) and 28 (red symbols) (AUBC_M_) observed in mono (rhombs) and combined therapy (squares) simulations. DOSE/MIC ratios were calculated using MICs determined at SI (**a**,**b**) and HI (**c**,**d**). The combination MICs were determined by method 1 (MIC1, (**a**,**c**)) and method 2 (MIC2, (**b**,**d**)). The relationships fit by Equation (1): (**c**) x0 = −1.175, a = 900, b = 0.266 and (**d**) x0 = −0.955, a = 900, b = 0.451.

**Table 1 antibiotics-12-01705-t001:** Modal MICs (mg/L) of single and combined with relebactam doripenem at SI and HI density against *K. pneumoniae*.

*K. pneumoniae* Isolate	Doripenem			Doripenem+ Fixed Relebactam Concentration			Doripenem+ Relebactam Fixed PK-Based Concentration Ratio 1.5/1			Doripenem+ Relebactam Fixed PK-Based Concentration Ratio 6/1			Doripenem+ Relebactam Fixed PK-Based Concentration Ratio 1/3		
	SI	HI	Fold MIC Increase	SI	HI	Fold MIC Increase	SI	HI	Fold MIC Increase	SI	HI	Fold MIC Increase	SI	HI	Fold MIC Increase
**28**	16	256	16 *	0.25	16	64 *	2	8	4	8	16	2	0.5	4	8 *
**19**	4	128	32 *	0.125	8	64 *	0.5	4	8 *	1	16	16 *	0.25	4	16 *

* MIC changes associated with IE.

## Data Availability

Data are contained within the article and Appendix A.

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
