# Peer review of "Pharmacodynamics of Doripenem Alone and in Combination with Relebactam in an In Vitro Hollow-Fiber Dynamic Model: Emergence of Resistance of Carbapenemase-Producing Klebsiella pneumoniae and the Inoculum Effect"

_antibiotics, 2023, doi:10.3390/antibiotics12121705_

Round 1
Reviewer 1 Report
Comments and Suggestions for Authors
The manuscript presents results about the inoculum effect and testing of resistance in the combination of doripenem and relebactam in an in vitro model using Klebsiella pneumonia. In my opinion, the topic is relevant in the field and addresses the public health problem of antibiotic resistance. Even though previous reports have addressed this issue, the in vitro model, the inclusion of simulation of data, and the combination of two different agents produce a publishable manuscript. The manuscript is well written, and in my consideration, the methodology is appropriate to test the hypothesis and conclude according to the objective of the research question. I suggest adding a scheme of the methodology to become a more attractive and citable article for readers.
Reviewer 2 Report
Comments and Suggestions for Authors
Reviewer’s Comments and Suggestions for Authors
Journal: Antibiotics, MDPI
Manuscript ID: antibiotics-2727464
Type: Article
Title: Pharmacodynamics of doripenem alone and in combination with relebactam in an in vitro hollow-fiber dynamic model: emergence of resistance of KPC-producing Klebsiella pneumoniae and the inoculum effect
Authors: Elena N. Strukova, Maria V. Golikova, Svetlana A. Dovzhenko, Mikhail B. Kobrin and Stephen H. Zinner
The authors of the manuscript antibiotics-2727464 investigated 5-day treatments with doripenem and its combination with relebactam at different ratios of the agents in a hollow-fiber dynamic model against two carbapenemase-producing K. pneumoniae 28, and 19 strains at standard and high inocula. Combination MICs were tested using traditional (fixed relebactam concentration) and a pharmacokinetic-based approach (fixed doripenem-to-relebactam concentration ratio equal to the therapeutic 24-h area under the concentration-time curve ratio). The results of this manuscript revealed that (1) therapeutic regimens of doripenem as well as a combination of doripenem with relebactam was not able to completely suppress the emergence of resistance; (2) the clinical relevance of the IE for carbapenems and their combinations with carbapenemase inhibitors; (3) the applicability of a pharmacokinetic-based approach to evaluate carbapenem MICs in the presence of an inhibitor to predict the IE and to describe the patterns of resistance occurrence.
The manuscript is written well. The authors have stated several limitations of the study. I will recommend this manuscript for publication in the journal Antibiotics.
Minor essential revisions
1. Abbreviations and acronyms are typically defined the first time the term is used in the abstract and in the main text, and then abbreviations are used throughout the remainder of the manuscript. Please consider adhering to this convention, and check throughout the manuscript. For example, MIC, AUC, CFU/mL.
2. Line 14: the word size should be consistent.
3. Line 38: in vitro.
4. Line 79: please describe the “both K. pneumoniae strains”.
5. Line 120: mg*L/h? please check throughout the manuscript.
6. Line 173: CFU/mL, please check throughout the manuscript.
7. Line 183: 4–8-fold, please check throughout the manuscript.
8. Line 224: method1.
9. Line 348: 5 × 105 CFU/mL, please check throughout the manuscript.
10. Line 352: 18 h, please check throughout the manuscript.
11. Lines 457-458: please rephrase the sentence.
12. Figures 1, 2, and 3: X axis: Time (h), please check all the figures.
13. Please format the references according to the guidance of the journal Antibiotics.
14. Please check English typing errors throughout the manuscript.
Comments on the Quality of English Language
Reviewer’s Comments and Suggestions for Authors
Journal: Antibiotics, MDPI
Manuscript ID: antibiotics-2727464
Type: Article
Title: Pharmacodynamics of doripenem alone and in combination with relebactam in an in vitro hollow-fiber dynamic model: emergence of resistance of KPC-producing Klebsiella pneumoniae and the inoculum effect
Authors: Elena N. Strukova, Maria V. Golikova, Svetlana A. Dovzhenko, Mikhail B. Kobrin and Stephen H. Zinner
The authors of the manuscript antibiotics-2727464 investigated 5-day treatments with doripenem and its combination with relebactam at different ratios of the agents in a hollow-fiber dynamic model against two carbapenemase-producing K. pneumoniae 28, and 19 strains at standard and high inocula. Combination MICs were tested using traditional (fixed relebactam concentration) and a pharmacokinetic-based approach (fixed doripenem-to-relebactam concentration ratio equal to the therapeutic 24-h area under the concentration-time curve ratio). The results of this manuscript revealed that (1) therapeutic regimens of doripenem as well as a combination of doripenem with relebactam was not able to completely suppress the emergence of resistance; (2) the clinical relevance of the IE for carbapenems and their combinations with carbapenemase inhibitors; (3) the applicability of a pharmacokinetic-based approach to evaluate carbapenem MICs in the presence of an inhibitor to predict the IE and to describe the patterns of resistance occurrence.
The manuscript is written well. The authors have stated several limitations of the study. I will recommend this manuscript for publication in the journal Antibiotics.
Minor essential revisions
1. Abbreviations and acronyms are typically defined the first time the term is used in the abstract and in the main text, and then abbreviations are used throughout the remainder of the manuscript. Please consider adhering to this convention, and check throughout the manuscript. For example, MIC, AUC, CFU/mL.
2. Line 14: the word size should be consistent.
3. Line 38: in vitro.
4. Line 79: please describe the “both K. pneumoniae strains”.
5. Line 120: mg*L/h? please check throughout the manuscript.
6. Line 173: CFU/mL, please check throughout the manuscript.
7. Line 183: 4–8-fold, please check throughout the manuscript.
8. Line 224: method1.
9. Line 348: 5 × 105 CFU/mL, please check throughout the manuscript.
10. Line 352: 18 h, please check throughout the manuscript.
11. Lines 457-458: please rephrase the sentence.
12. Figures 1, 2, and 3: X axis: Time (h), please check all the figures.
13. Please format the references according to the guidance of the journal Antibiotics.
14. Please check English typing errors throughout the manuscript.
Reviewer 3 Report
Comments and Suggestions for Authors
Strukova and co-authors made interesting observations on the inoculum effect and the emergence of antibiotic resistance in Klebsiella pneumoniae using the hollow-fiber infection model. The authors clearly discuss the differences in results obtained from traditional methods for MIC determination versus the HFIM. The authors first determine the MIC of doripenem independently, and in combination with relebactam at low population density (SI) as well as high population density (HI). Next, they simulate the pharmacokinetics and dynamics of doripenem/relebactam at different combination ratios. They perform time kill experiments to study the emergence of resistant strains after antibiotic treatment, and also study DOSE/MIC relationships as means to predict MICs and resistance emergence.
The study is thorough and offers the readers new and clinically important information to the scientific and medical community. Therefore, I recommend publishing the manuscript in Antibiotics. The manuscript is well constructed, although may be edited at specific places reduce the word count and improve the clarity for communication.
Here are my minor comments:
1. L69-L70: Authors present their secondary goal of the study here. They may take this opportunity to discuss specifically how the studies on IE will affect clinical doses. For instance, will the doses change according the stage of infection? Or whether dose-adjustment is necessary as the bacterial numbers will vary between patients? Please cite papers accordingly, if this advice is considered.
2. Table 1: By comparing the ‘fold MIC increase’, it seems that the IR is higher when the experiment is done using method 1 (64 fold). Alternatively, it is lower when the studies are performed using method 2 (<20 fold). Under the discussion section, authors may clearly discuss if these differences manifest due to the experimental set up.
3. L277-L278 and Figure 4 (a and b): Data may be fitted if there were more data points.
